# Synthesis of CoFe_2_O_4_/Peanut Shell Powder Composites and the Associated Magnetic Solid Phase Extraction of Phenoxy Carboxylic Acid Herbicides in Water

**DOI:** 10.3390/ijerph19148450

**Published:** 2022-07-11

**Authors:** Dongliang Ji, Zhaoqin Huang, Buyun Du

**Affiliations:** College of Environment and Ecology, Jiangsu Open University, Nanjing 210036, China; jidl@jsou.edu.cn (D.J.); duby@jsou.edu.cn (B.D.)

**Keywords:** magnetic biochar material, phenoxy carboxylic acid herbicides, magnetic solid phase extraction, actual water sample

## Abstract

The magnetic biochar material CoFe_2_O_4_/PCPS (peanut shell powder) was prepared based on the hybrid calcination method. The properties of prepared composites and the extraction effect of magnetic solid phase extraction on phenoxy carboxylic acid herbicides were assessed. The morphology, crystal structure, specific surface area, and pore size distribution of the material were analysed using a transmission electron microscope (TEM), infrared Fourier transform infrared spectroscopy (FTIR), X-ray powder diffraction (XRD), X-ray photoelectron spectroscopy (XPS), and N_2_ absorption surface analysis (BET). The results of the magnetic solid phase extraction of a variety of phenoxy carboxylic acid herbicides in water using CoFe_2_O_4_/PCPS composites showed that, when the mass ratio of CoFe_2_O_4_ and PCPS was 1:1, 40 mg of the composite was used, and the adsorption time was 10 min at pH 8.50. Methanol was used as the eluent, and the recovery rates of the three phenoxy carboxylic acid herbicides were maintained at 81.95–99.07%. Furthermore, the actual water sample analysis results showed that the established method had good accuracy, stability, and reliability.

## 1. Introduction

Phenoxy carboxylic acid herbicides are widely used due to their low cost, effectiveness in weed control, and high water solubility [1,2]. However, they still have the potential to cause harm to animal and human populations, even though their concentrations in environmental water samples are relatively low. Long-term exposure will increase the risks of leukemia and non-Hodgkin lymphoma in children, endocrine disorders in organisms, and metabolic imbalance [3,4,5,6]. Therefore, it is of theoretical and practical significance to develop a method for detecting trace amounts of phenoxy carboxylic acid herbicides.

As a new technology for sample pretreatment, magnetic solid phase extraction (MSPE) can fully contact the target analyte and quickly enrich the analyte. The magnetic sorbents can be directly recycled by a magnet [7]. Consequently, MSPE has a high separation efficiency and convenient operation [8]. The key to the MSPE is the choice of sorbent. Generally, graphene oxide (GO), metal–organic frameworks (MOFs), covalent organic frameworks (COFs), carbon nanotubes (CNTs), and biochar (BC) may be used as sorbents. Biochar with a large specific surface area enables the efficient extraction of the target analyte from water [9]. In this work, the preparation of CoFe_2_O_4_/PCPS (peanut shell powder) and its physicochemical properties have been characterized. The magnetic biochar material CoFe_2_O_4_/PCPS was synthesized with peanut shell as the carbon source and CoFe_2_O_4_ as the magnetic nucleus. The CoFe_2_O_4_ had a face-centered cube structure formed by Fe^3+^, Co^2+^, and O^2−^. Three substances, including 4-chlorophenoxyacetic acid, 2,4-dichloropheno-xyacetic acid, and 2,4-droplet propionic acid, were chosen as the target analytes, then the effects of preparation conditions and external factors on the recovery of analytes were discussed. Finally, a pretreatment method to concentrate the phenoxy carboxylic acid herbicides and their subsequent analysis was established.

## 2. Materials and Methods

### 2.1. Preparation of CoFe_2_O_4_ Nanoparticles

A total of 0.01 mol Co(NO_3_)_2_·6H_2_O and 0.02 mol FeCl_3_·6H_2_O were added to beakers with 30 mL pure water. The pH was adjusted to 12 using a 30% NaOH solution. The mixture was stirred and sonicated for 1 h. Subsequently, the mixture was transferred to a vacuum drying oven and reacted at 120 °C for 10 h. The product was washed, dried, and stored for subsequent use.

### 2.2. Preparation of CoFe_2_O_4_/PCPS Composites

The fresh peanut shells were washed and oven-dried at 110 °C for 24 h, crushed into powder by a grinder, and then sieved. Particles with a diameter of 0.25–0.425 mm were collected. CoFe_2_O_4_ (1.0000 g), peanut shell powder (PCPS, 1.0000 g), and KCl (2.0000 g) were placed in a mortar and ground evenly. After this, the CoFe_2_O_4_, peanut shell powder (PCPS), and KCl mixture was stirred with 20 mL water for 24 h and oven dried at 120 °C for 12 h. Subsequently, the material was placed into a tube furnace with an argon atmosphere and calcined at 600 °C for 3 h. Finally, the sample was cleaned, dried, and named CS1 for subsequent use. Similarly, the materials with mass ratios of CoFe_2_O_4_ and PCPS of 1:0.5, 1:1.5, and 1:2 were named CS0.5, CS1.5, and CS2, respectively.

### 2.3. Collection of Environmental Water

Three river water samples were randomly gathered from the Yellow and Yangtze Rivers in addition to water samples from the Xuanwu Lake in Nanjing. The snow water was taken from the school campus. All the water samples were stored in the dark at low temperature. The samples were cleaned preliminarily through filtration using a 0.45 µm membrane.

### 2.4. Magnetic Solid Phase Extraction Experiment

A total of 40 mg magnetic nanocomposite was added into a 100 mL water sample and treated with ultrasonic waves for 10 min so that the mixture attained equilibrium of adsorption and desorption. Under the action of an external magnetic field, the water sample and the composite material were separated. Subsequently, 2 mL methanol was used to elute the analyte absorbed by the composite material. The eluent was analysed, and the analyte was detected by high performance liquid chromatography (HPLC). The material synthesis and MSPE process are shown in Figure 1. HPLC instrument configuration and chromatographic conditions are shown in Table 1. The standard curve is shown in Table 2.

### 2.5. Analysis Methods

A transmission electron microscope (TEM) (JEM-200 CX, JEOL, Tokyo, Japan) and scanning electron microscopy (SEM) (S-3400N Ⅱ, Hitachi, Tokyo, Japan) were used to examine the morphological and chemical properties for the CoFe_2_O_4_/PCPS. A spectrometer (Tensor 27, Bruck, Germany) was used to obtain the infrared Fourier transform spectroscopy (FTIR) spectra of the CoFe_2_O_4_/PCPS. The samples were also analysed using X-ray photoelectron spectroscopy (XPS) (PHI 5000 Versa Probe, Ulvac-Phi, Chigasaki, Japan) and X-ray powder diffraction (XRD) (XRD-6000, Shimadzu, Tokyo, Japan). N_2_ absorption surface analysis was used to characterize the specific surface area and pore size distribution (ASAP2050, Micromertics, Norcross, GA, USA). Chromatographic analysis was recorded on a Waters HPLC (Waters 2489, Waters, Milford, MA, USA).

### 2.6. The Determination of Zete Potential

The samples were added to 50 mL purified water at a solid–liquid ratio of 1:1. After ultrasonic dispersion, the pH of the system was adjusted with 0.1 mol·L^−1^ HCL and NaOH. The potential of supernatant was determined by Zeta potentiometer. Each sample was measured three times.

## 3. Results

### 3.1. Morphology Analysis

The material surface morphologies are shown in Figure 2, in which Figure 2A–C displays the transmission electron microscope (TEM), high-resolution transmission electron microscopy (HRTEM), and selected area electron diffraction (SAED) images of CoFe_2_O_4_, respectively. Figure 2F displays the TEM and HRTEM images of PCPS. Finally, Figure 2G–I shows the TEM, HRTEM, and SAED images of CoFe_2_O_4_/PCPS, respectively.

As shown in Figure 2A–C, the CoFe_2_O_4_ was composed of homogeneous nanoparticles with diameters of 10–20 nm. The lattice planes with separation distances of 0.4882, 0.2906, 0.2573, and 0.1282 nm corresponded to the (111), (220), (311), and (533) CoFe_2_O_4_ lattice planes, respectively [10,11]. Figure 2D,E exhibited the presence of carbon quantum dots that were distributed uniformly within the size interval of 1.3–2.5 nm. Two groups of lattice fringes in Figure 2F represented the crystal planes (100) and (101) for graphite carbon [12,13,14]. Figure 2G exhibited that many holes were distributed on the PCPS lamellar structure. The CoFe_2_O_4_ lattice-stripe nanoparticles appeared in Figure 2H,I. This indicated that CoFe_2_O_4_ nanoparticles had been successfully compounded with PCPS.

### 3.2. XRD Analysis

Figure 3A–C represents the typical XRD pattern for PCPS, CoFe_2_O_4_, and CoFe_2_O_4_/PCPS XRD, respectively.

Figure 3A shows broad peaks at 24.0°, which were indexed as the (002) graphite carbon planes [15,16,17,18,19]. Nine discernible diffraction peaks are observed in Figure 3B, including those at 18.2°, 30.1°, 35.5°, 43.5°, 53.9°, 57.2°, 62.7°, 65.7°, and 74.0°. They can be ascribed to the (111), (220), (311), (400), (422), (511), (440), (531), and (533) planes, respectively [20]. Figure 3C shows that the diffraction peaks of CoFe_2_O_4_ were diminished due to the covering PCPS; however, the crystal shape did not change.

### 3.3. FTIR Analysis

The infrared spectra of CoFe_2_O_4_ (A), PCPS (B), and CoFe_2_O_4_/PCPS (C) are shown in Figure 4.

The band at approximately 3405 cm^−1^ represented the O-H stretching vibration [21] and is shown in Figure 4A–C. The bands at 1635 cm^−1^, 1384 cm^−1^, and 400–650 cm^−1^ were assigned to the O-H, NO^3−^, Fe-O, and Co-O groups [22], respectively. These bands are shown in Figure 4A. The bands at 1588 cm^−1^, 1379 cm^−1^, and 980–1200 cm^−1^ were attributed to the C=C, -COOH, C-O, and OH groups [23,24], respectively. As shown in Figure 4C, all the generally observed peaks for CoFe_2_O_4_ and PCPS were present in the synthesized materials, but their spectrum intensities decreased slightly. All the results indicate that the composite possessed oxygen-containing and carbon-containing functional groups on the surface. These groups can adsorb target molecules by hydrophobic interaction and hydrogen bonding [25].

### 3.4. XPS Analysis

Figure 5 shows the XPS spectra in the prepared CoFe_2_O_4_/PCPS composite. The spectra for C, O, Fe, and Co along with their chemical bonding states are shown in Figure 5A–D, respectively. The illustration shown in Figure 5A displays the full spectrum of these elements.

As shown in Figure 5A, the C 1 s, O 1s, Fe 2p, and Co 2p binding energies were 285.9 eV, 529.4 eV, 714.0 eV, and 782.8 eV, respectively. The atomic ratio for Co and Fe was 1:2, which corresponds with the atom number ratio of Co and Fe in CoFe_2_O_4_. The binding energies 284.5 eV, 284.8 eV, 286.1 eV, and 288.8 eV were attributed to the C=C, C-H, C-O, and (C=O)OH functional groups. The binding energies of 530.3 eV, 531.0 eV, 532.0 eV, and 533.2 eV in Figure 5B were attributed to lattice oxygen, Fe-O-C, C(O)OH, and C-OH, respectively. In addition, the binding energies shown in Figure 5C,D—712.1 eV, 725.8 eV, 781.2 eV, and 782.6 eV—were attributed to the Fe 2p3/2 and Fe 2p1/2 octahedron along with the Co^2+^ tetrahedron, respectively [26,27].

### 3.5. N_2_ Adsorption-Desorption Analysis

Figure 6A–C represents the nitrogen adsorption–desorption curves for CoFe_2_O_4_, PCPS and CoFe_2_O_4_/PCPS, respectively. The illustrations in each figure displayed their pore size distribution, which was also shown in the figure panels.

As shown in Figure 6, the N_2_ adsorption–desorption isotherms for CoFe_2_O_4_ conformed to type III, while the PCPS isotherms and the CoFe_2_O_4_/PCPS conformed to type IV. The pore sizes of CoFe_2_O_4_, PCPS, and CoFe_2_O_4_/PCPS were 9.7–20.3 nm, 0.6–2.0 nm, and 1.8–20.5 nm, respectively. This indicated that CoFe_2_O_4_/PCPS was microporous and mesoporous materials. In addition, the specific surface areas of CoFe_2_O_4_, PCPS, and CoFe_2_O_4_/PCPS were 4.64, 152.90, and 146.03 m^2^·g^−1^, respectively, which indicated that PCPS still had a large specific surface area after covering CoFe_2_O_4_.

### 3.6. MSPE Condition Optimizing

#### 3.6.1. Material Components

Magnetic solid phase extraction was performed using the prepared materials to discuss the influence of material components on extraction effects. For each treatment, 40 mg CS0.5, CS1, CS1.5, and CS2 composite materials were added into four samples containing 100 mL analyte (4-PA, 2,4-D, 2,4-DP) at concentrations of 0.04 mg·L^−1^ each. Subsequently, they were eluted by 2 mL acetonitrile. The results were shown in Figure 7.

As shown in Figure 7, the CS1 composite possessed the best magnetic solid phase extraction effect for three analytes. The principal reason was that when the carbon content was lower, the material had fewer adsorption sites. This was not conducive to adsorption of the analyte on the material surface, resulting in a relatively low recovery rate. When the carbon content increased, the magnetic properties of the composite diminished, and the composites were difficult to separate from the water sample under external magnetic fields. Therefore, the CS1 material with a CoFe_2_O_4_ and PCPS mass ratio of 1:1 was selected in the following experiments.

#### 3.6.2. Conditional Experiment

The effects of composite dosage, pH, eluent type, adsorption time, and enrichment factor on the pretreatment of the target analyte in the magnetic solid phase extraction process are shown in Figure 8.

Figure 8A shows that the recovery rates of the analytes increased with increasing dosages of CS1. When the SC1 dosage reached 40 mg, the recovery rate of the analytes exceeded 82.17%, but as the usage of CS1 continued to increase, the recoveries of the three analytes remained between 81.17% and 99.77%. However, the elution process in the subsequent treatment became more difficult. Therefore, 40 mg should be selected as the appropriate dosage. From the results of Figure 8B, we found that the target analyte had the optimum recovery at a pH of 8.50. A possible cause is that the surface of the material was negatively charged in the weakly alkaline system (Figure 8C). The carboxylate ions dissociated from the target pollutants may have combined with the material through the electrostatic effect of NH_4_^+^ in the buffer solution; this is beneficial to the adsorption process. As Figure 8D shows, the magnetic solid phase extraction effect was optimal when methanol was used as the eluent and the recovery rate of phenoxy carboxylic acid herbicides was between 82.17 and 99.07%. Consequently, methanol was used as the eluent in the magnetic solid phase extraction experiment.

The speed of magnetic solid phase extraction determined the viability of this method for practical application. As shown in Figure 8E, the recoveries of the three phenoxy carboxylic herbicides increased initially and subsequently stabilized. When the adsorption time was less than eight minutes, the recoveries of the analytes were less than 55.72%. When the adsorption time was greater than ten minutes, the recovery rate was between 81.95% and 99.07%. Equilibrium between adsorption and desorption was observed. The enrichment factor condition experiment from Figure 8F showed that when the enrichment factor was greater than 75, the recovery rates of the three phenoxy carboxylic herbicides were all less than 80%. This was because increasing the sample volume reduced the mass of material contained per sample volume; consequently, the analyte could not be completely adsorbed. Therefore, 50 was determined as the enrichment factor in the magnetic solid phase extraction experiment. According to Figure 8G, the recovery of SC1 does not decrease significantly after 10 recycles.

### 3.7. Analysis of Environmental Water Samples

To verify the reliability of the established method, the MSPE method was used to analyse actual water samples from the Yellow and Yangtze Rivers, Xuanwu Lake, and snow melt. None of the collected water samples were contaminated by 4-PA, 2,4-D, or 2,4-DP. The water samples were spiked with 0.02 mg·L^−1^ and 0.04 mg·L^−1^ of each analyte. Based on magnetic solid phase extraction and high-performance liquid chromatography, the analyte recoveries varied from 75.22% to 94.66%, and the relative standard deviation (RSD) ranged from 0.08% to 3.70%, as shown in Table 3. Comparing the method established in this paper with other reported methods, as shown in Table 4 [1,28,29,30,31], the results of the current study showed that the pretreatment method based on CoFe_2_O_4_/PCPS magnetic biochar material had good accuracy, stability, and reliability.

## 4. Conclusions

In this paper, a CoFe_2_O_4_/PCPS composite was prepared via the biopyrolysis method and applied to the magnetic solid phase extraction of three phenoxy carboxylic herbicides (4-PA, 2,4-D, 2,4-DP) dissolved in water. The influence of material composition, dosage, pH, adsorption time, eluent, enrichment factor, and cycles on the magnetic solid phase extraction effect was discussed. The optimal conditions were applied for the actual detection of the herbicides in environmental water samples, and the main conclusions were as follows:

(1)The characterizations such as TEM, XRD, FTIR, and BET demonstrated that the CoFe_2_O_4_/PCPS composite was successfully synthesized. The material maintained the original structure of PCPS and had a large specific surface area and pore volume.(2)In the optimization experiment of magnetic solid phase extraction conditions, phenoxy carboxylic herbicides were extracted using 40 mg of the composite material, and adsorption occurred in water samples with a pH of 8.50 for 10 min. The best magnetic solid phase extraction effect was obtained with methanol as the eluent. The recovery rates of all the analyte were greater 82.17%. The material still maintains high performance after ten cycles.(3)When the established method was applied to environmental water samples, the recoveries were 75.22–94.66%. The RSD was 0.08–3.70%.

## Figures and Tables

**Figure 1 ijerph-19-08450-f001:**
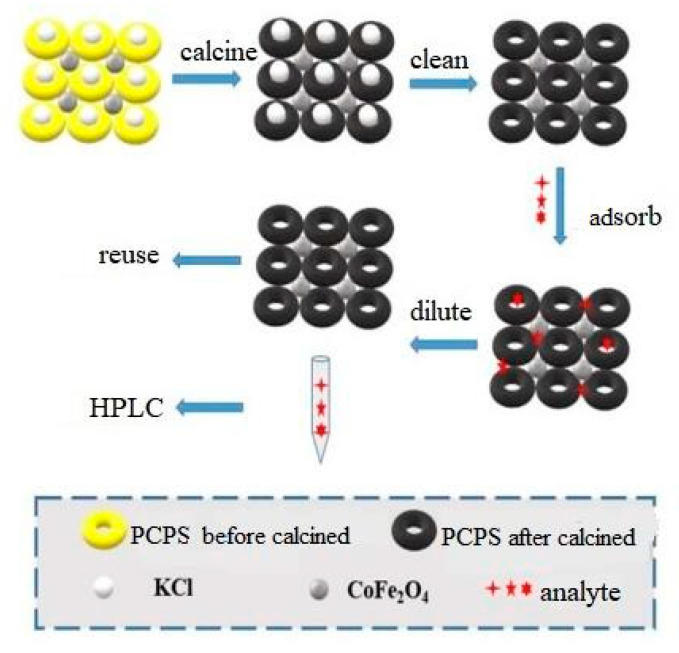
The schematic diagram of material synthesis and MSPE.

**Figure 2 ijerph-19-08450-f002:**
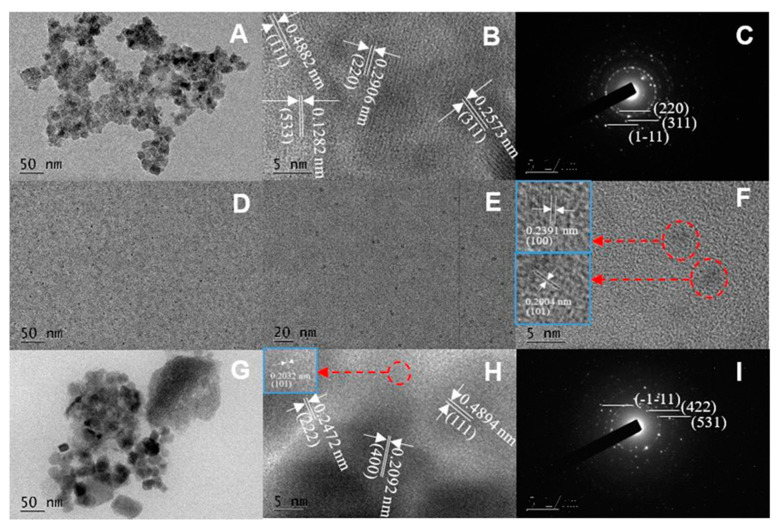
The TEM, HRTEM, and SAED images of CoFe_2_O_4_ (**A**–**C**); the TEM images of PCPS (**D**,**E**) and HRTEM (**F**); the TEM, HRTEM, and SAED images of CoFe_2_O_4_/PCPS (**G**–**I**).

**Figure 3 ijerph-19-08450-f003:**
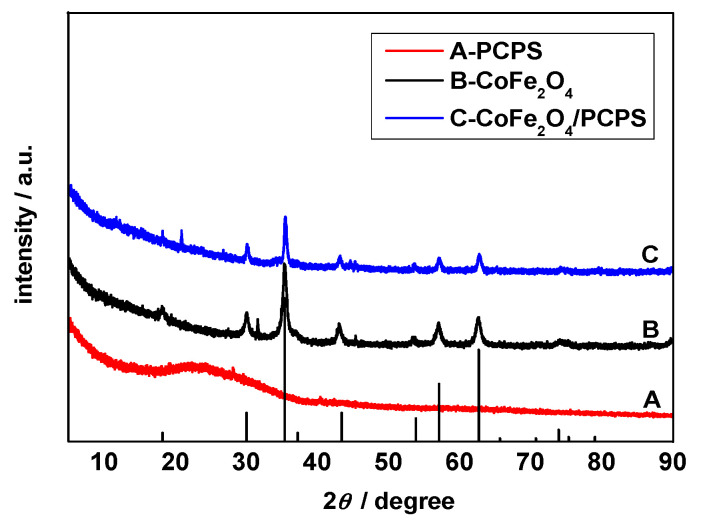
XRD patterns of synthesized PCPS (**A**), CoFe_2_O_4_ (**B**), and CoFe_2_O_4_/PCPS (**C**).

**Figure 4 ijerph-19-08450-f004:**
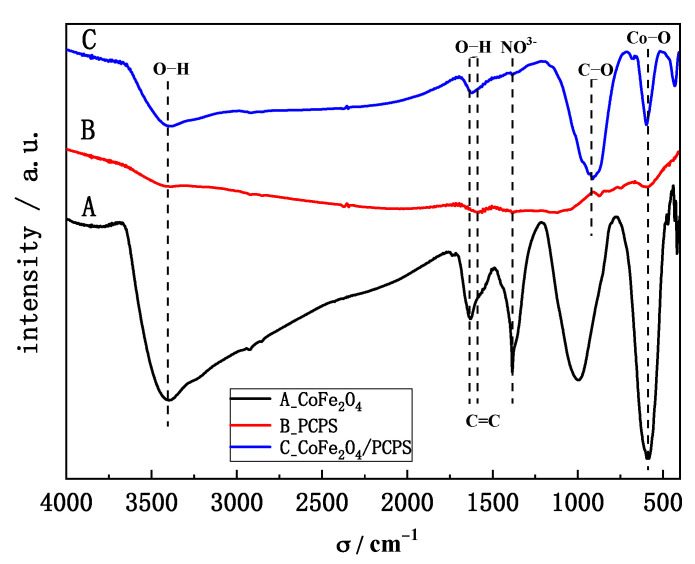
FTIR spectra for synthesized CoFe_2_O_4_ (**A**), PCPS (**B**), and CoFe_2_O_4_/PCPS (**C**).

**Figure 5 ijerph-19-08450-f005:**
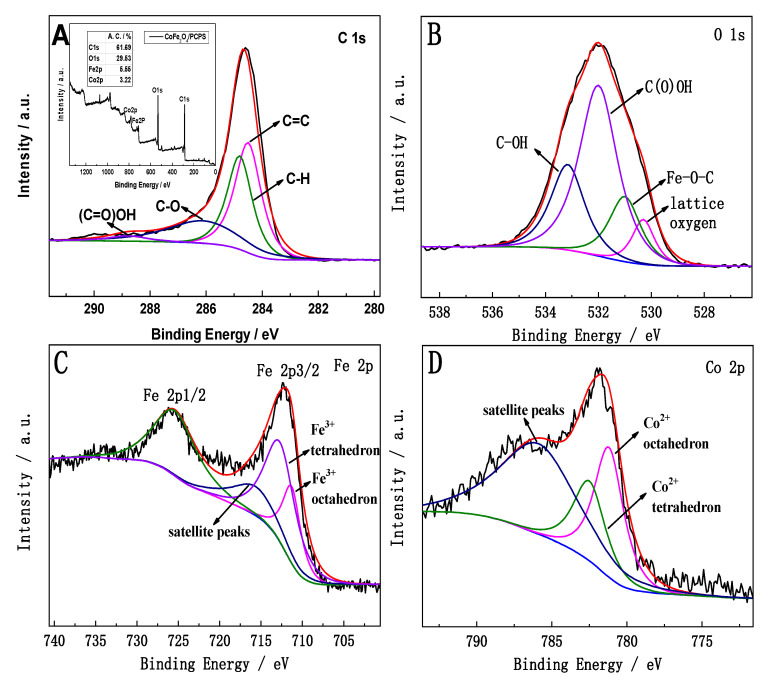
XPS for CoFe_2_O_4_/PCPS: full scan spectra [the inset in (**A**)], C 1s spectra (**A**), O 1s spectra (**B**), Fe 2p spectra (**C**), and Co 2p spectra (**D**).

**Figure 6 ijerph-19-08450-f006:**
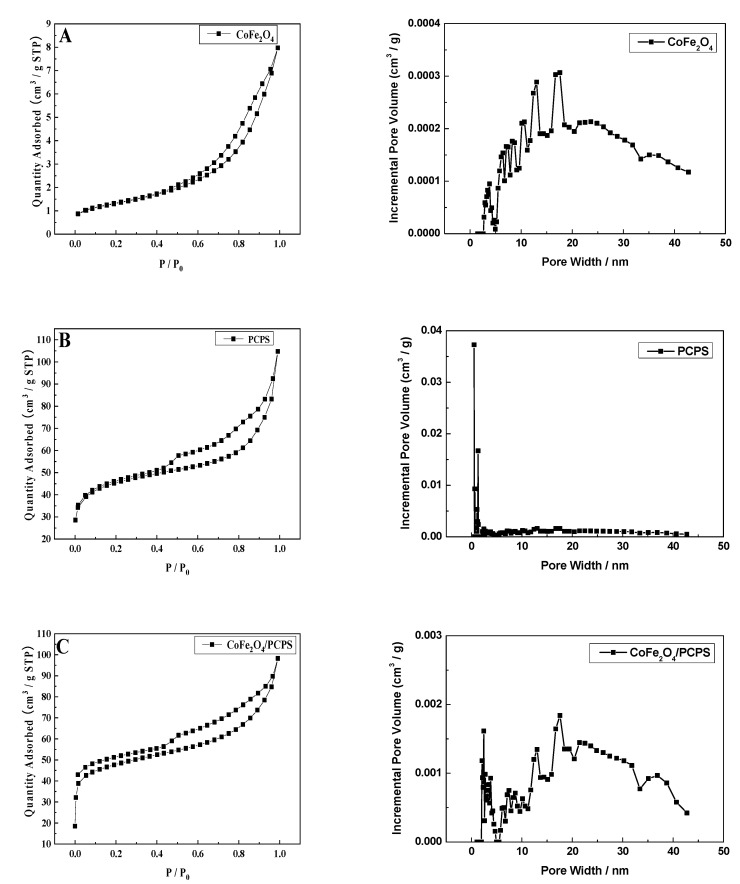
N_2_ adsorption–desorption isotherms and the pore size distribution of the CoFe_2_O_4_ (**A**), PCPS (**B**), and CoFe_2_O_4_/PCPS (**C**).

**Figure 7 ijerph-19-08450-f007:**
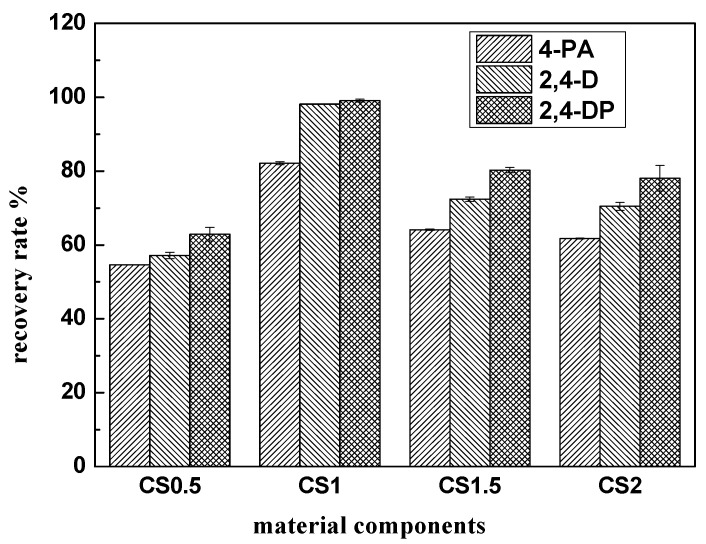
The effect of material composition on the magnetic solid phase extraction.

**Figure 8 ijerph-19-08450-f008:**
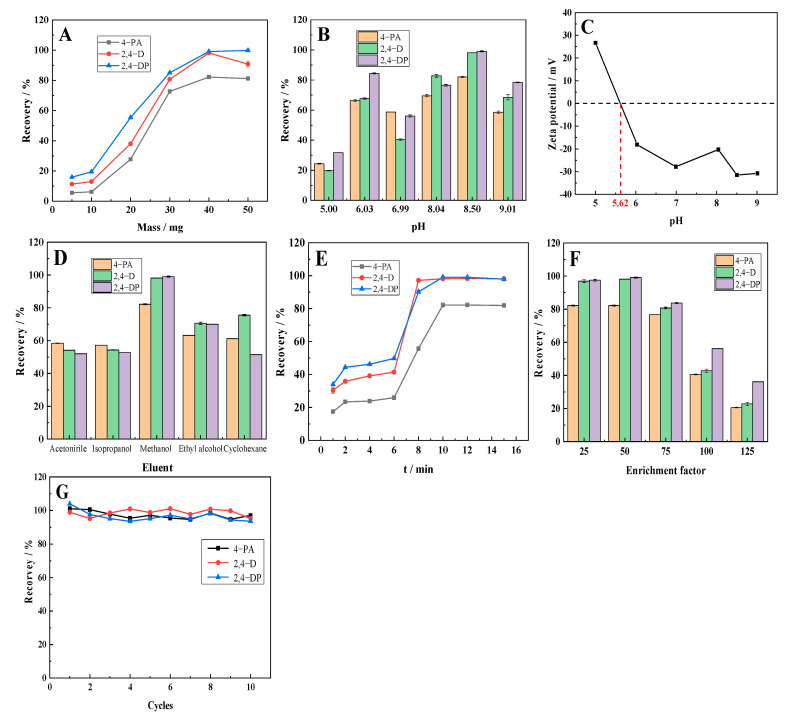
The effect of dosage (**A**), pH (**B**), Zeta potential (**C**), eluent type (**D**), adsorption time (**E**), enrichment factor (**F**), and cycles (**G**) on three phenoxy carboxylic acid herbicides.

**Table 1 ijerph-19-08450-t001:** The HPLC instrument configuration and chromatographic conditions.

Item	Parameter
Instruments	Waters 2489
Detector	An UV/Visible detector
Chromatographic column	A Lichrospher C18 column (150 mm × 4.6 mm, 5 μm)
Detection wavelength	230 nm
Mobile phase (volume ratio)	Methanol to water to acetic acid = 50:50:0.2
Flow rate	1.0 mL·min^−1^
Injection volume	20.0 µL
Column temperature	30 °C

**Table 2 ijerph-19-08450-t002:** Analytical parameters of MSPE-HPLC-UV method for the determination of phenoxy carboxylic acid pesticides in standard solution.

Analyte	Standard Curve	Linear Range (μg·L^−1^)	Correlation Coefficient (r^2^)	*LOD* (μg·L^−1^)	*LOQ* (μg·L^−1^)
4-PA	*y* = 60.4608*x* + 240.4391	1.0–1000	0.9996	0.30	0.98
2,4-D	*y* = 37.6800*x* + 984.8045	2.0–1000	0.9978	0.58	1.94
2,4-DP	*y* = 35.9193*x* − 832.7198	2.0–1000	0.9995	0.59	1.96

**Table 3 ijerph-19-08450-t003:** The recovery and RSD of the three analytes in real water samples (na = 3).

Water Samples	Standard Addition Concentration (mg·L^−1^)	Detection of Concentration (mg·L^−1^)			Recovery Rate ^b^/% (*RSD*/%)		
		4-PA	2,4-D	2,4-DP	4-PA	2,4-D	2,4-DP
Yellow River	0	——	——	——			
	0.02	0.0164	0.0175	0.0189	81.75 (1.66)	87.45 (2.06)	94.66 (3.15)
	0.04	0.0333	0.0346	0.0355	83.27 (2.39)	86.41 (1.52)	88.63 (1.99)
Xuanwu Lake	0	——	——	——			
	0.02	0.0150	0.0159	0.0177	75.22 (1.25)	79.66 (3.35)	88.29 (3.70)
	0.04	0.0335	0.0333	0.0347	83.82 (0.85)	83.13 (3.15)	86.70 (2.97)
Yangtze River	0	——	——	——			
	0.02	0.0161	0.0163	0.0163	80.27 (0.61)	81.65 (1.89)	81.50 (1.25)
	0.04	0.0336	0.0353	0.0370	84.04 (2.27)	88.15 (1.62)	92.61 (0.08)
Snow melt	0	——	——	——			
	0.02	0.0155	0.0161	0.0162	77.30 (1.48)	80.36 (1.38)	81.08 (0.56)
	0.04	0.0328	0.0339	0.0358	81.88 (2.62)	84.86 (1.73)	89.54 (1.46)

Detection times; ^b^: average value of three parallel experiments.

**Table 4 ijerph-19-08450-t004:** Method comparison.

Method	Analytical Sample	Sample Volume /mL	Pre-Conditioning Time/min	*LOD*s (μg·L^−1^)			*RSD*/%	Recovery Rate/%	Reference
4-PA	2,4-D	2,4-DP
DSPE ^a^	tap water and lake water	50	13	0.2–0.3			1.4–8.6	83.7–114.4	[1]
SPE	river water and waste water	750	11			0.3–6.3	1.1–11.4	95–104	[28]
SPE	distilled water, stream and well water	200	100		0.02		0.2–4.0	80.0–110.0	[29]
SPE	river water	400	90		0.01			50.0–80.0	[30]
μ-SPE ^b^-MSPE	Reservoir raw water	10	45		0.2		1.7–5.1	89.0–103.0	[31]
MSPE	river water, lake water and snow water	100	10	0.3	0.58	0.59	0.08–3.70	75.22–94.66	this text

^a^: Dispersed solid phase extraction; ^b^: micro solid phase extraction.

## Data Availability

Not applicable.

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
