# Peer review of "Synthesis of CoFe2O4/Peanut Shell Powder Composites and the Associated Magnetic Solid Phase Extraction of Phenoxy Carboxylic Acid Herbicides in Water"

_ijerph, 2022, doi:10.3390/ijerph19148450_

Round 1

Reviewer 1 Report

Section 2.4. The way the HPLC parameters are presented is odd. These data could be included in a table or be part of the text.

Section 3.1. The acronyms "TEM, HRTEM, and SAED" have not been defined.

Section 3.3. The sentence "These groups can adsorb target molecules by hydrophobic interaction and hydrogen bonding." needs a reference.

Figure 8. Chinese characters should be translated into English.

Section 3.6.2 on Page 8, last line. Please explain the meaning of "enrichment factor", and I guess that 8(E) should be 8(F). The sentence "This was because increasing the sample volume reduced the mass of material contained per sample volume; consequently, the analyte could not be completely adsorbed" is not understandable.

Figure 9 is meaningless. Please remove or justify.

Conclusions. I have not seen the reusability experiments. Please clarify.

Author Response

Thanks very much for the comment.

Reviewer 2 Report

There are several problems with this manuscript in its present form. XRD characterization of the materials needs to be improved.  Figure 3(A) shows broad peaks at 24.0°,  and the authors have  indexed it  as the (002) graphite carbon planes. It is not possible to index such crsytallographic plane ( Bragg reflection) with such low quality XRD data. There are no references concerning the atribution of the Bragg reflections of the as made materials. SEM and TEM  figures are very overlapped and it is very difficult to analyze them. Some of them are not even similar to the ones published currently in the literature.  FT-IR data and the correct vibration bands need to be assigned and discussed. Concerning the analysis of the water and the detection of the organic hazardous ( herbecides) using HPLC the authors need to provide more information about the experimental conditions of these analyses in order to be sure that the level of detection is worth of credit or not. In fact there is a lack of a study control in this study, since before using river water samples the authors could test their materials with more simple samples. The authors have provided data of zeta potential of their samples, nevertheless there is no description of these experiments in the manuscript. Overall, I would not advise the  publication of the manuscript in its present form.

Author Response

Thanks very much for the comment.

Reviewer 3 Report

The article entitled »Synthesis of CoFe2O4 / peanut shell powder composites and the associated magnetic solid phase extraction of phenoxy carboxylic acid herbicides in water« is interesting and allows the use of waste material for the adsorption of herbicides that can occur in waters. Analytical methods are suitable and can be used to precisely analyse the structural properties of materials. The article lacks of the economic component of this study, as the composite is planned to be used for the analysis of river waters and their pollution.

Note: Figure 6 is too small, and thus, unclear.

Author Response

Thanks very much for the comment.

Round 2

Reviewer 2 Report

The quality of the manuscript was  enhanced by the modifications made by  the authors. 

Author Response

Thanks for the suggestion. We have modified English in the abstract, introduction and conclusion of our manuscript carefully.